# Microstructure Characterisation and Identification of the Mechanical and Functional Properties of a New PMMA-ZnO Composite

**DOI:** 10.3390/ma13122717

**Published:** 2020-06-15

**Authors:** Rebeka Rudolf, Danica Popović, Sergej Tomić, Rajko Bobovnik, Vojkan Lazić, Peter Majerič, Ivan Anžel, Miodrag Čolić

**Affiliations:** 1Faculty of Mechanical Engineering, University of Maribor, Smetanova ulica 17, 2000 Maribor, Slovenia; peter.majeric@um.si (P.M.); ivan.anzel@um.si (I.A.); 2Zlatarna Celje d. o. o., Kersnikova 19, 3000 Celje, Slovenia; 3School of Dental Medicine, University of Belgrade, dr. Subotića 8, 11000 Belgrade, Serbia; danica.popovic@stomf.bg.ac.rs (D.P.); Vojkan.Lazic@stomf.bg.ac.rs (V.L.); 4Institute for the Application of Nuclear Energy, University of Belgrade, Banatska 31b, 11000 Belgrade, Serbia; sergej.tomic@gmail.com; 5Faculty of Polymer Technology, Ozare 19, 2380 Slovenj Gradec, Slovenia; rajko.bobovnik@ftpo.eu; 6Military Medical Academy Belgrade, Crnotravska 17, 11002 Beograd, Serbia; mjcolic@eunet.rs; 7Medical Faculty in Foca, University of East Sarajevo, Studentska 5, 73300 Foca, Republic of Srpska

**Keywords:** poly(methyl methacrylate) (PMMA), zinc oxide nanoparticles (ZnO NPs), composite, microstructure, characterisation, mechanical and functional properties, cytotoxicity, L929 cell cultures

## Abstract

In this research work, we synthesised poly(methyl methacrylate) (PMMA) enriched with 2 wt.% zinc oxide nanoparticles (ZnO) through conventional heat polymerisation and characterised its microstructure. It was found that the distribution of ZnO nanoparticles was homogeneous through the volume of the PMMA. The mechanical testing of the PMMA-ZnO composite primarily included the determination of the compressive properties on real dentures, while density measurements were performed using a pycnometer. The testing of functional properties involved the identification of the colour of the new PMMA-ZnO composite, where pure PMMA acted as a control. In the second step, the PMMA-ZnO cytotoxicity assays were measured in vitro, which were shown to be similar to the control PMMA. Based on this, it could be concluded that the newly formed PMMA-ZnO composite did not induce direct or indirect cytotoxic effects in L929 cell cultures; therefore, according to ISO/DIN 10993-5:2009, this composite was categorised as non-cytotoxic.

## 1. Introduction

Poly(methyl methacrylate) (PMMA), commercially available since 1937 and known as acrylic or acrylic glass, is a transparent thermoplastic and versatile material that has been used in a wide range of fields and applications [1,2]. PMMA is extensively applicable in medical and dental applications where purity and stability are critical for performance [3,4]. In dentistry, PMMA has an important role as a material for making complete or partial dentures. Namely, PMMA has a good degree of compatibility with human tissue, a fact that was discovered by the English Ophthalmologist Sir Harold Ridley in World War II. Royal Air Force pilots, whose eyes had been riddled with PMMA splinters coming from the side windows of their Supermarine Spitfire fighters, found that the plastic scarcely caused any rejection compared to the glass splinters coming from aircraft such as the Hawker Hurricane [5]. 

The most common problems with complete or partial dentures made using PMMA in the therapy of toothless people are bad mechanical properties and base cracking. All this requires frequent repairs, such as the production of the metal reinforcement grid, rebasing, or replacing them with a new one, which represents an additional financial burden for toothless patients [6,7,8].

Over the last few decades, the term nanotechnology, which refers to technology that involves nanosized particles where at least one of the dimensions of the material lie between 1 and 100 nm, has been established, and a new fast-growing market was born. The study of composite materials based on polymers and inorganic fillers of nanometric dimensions during the last few decades has developed into one of the most important industries in the field of nanoscience [9,10,11,12]. One of the promising candidates from the group of nanoparticles (NPs) in the form of metal oxides is ZnO, which is a largely inert white compound. Its various properties are dependent on the NPs’ size and shape, and therefore, they are marked as the crucial parameters for the synthesis and application in the different composite systems. Furthermore, ZnO exhibits antifungal and antimicrobial properties against both Gram-negative and Gram-positive bacteria. Results have indicated that ZnO NPs show antibacterial activity greater than microparticles do [13]. Furthermore, it was already demonstrated that the use of ZnO NPs increases the antimicrobial activity of some products, including against biofilms of *Staphylococcus aureus* and *Enterococcus faecalis* [14,15,16,17]. Accordingly, ZnO is used very widely from the chemical industry through to cosmetic and medical products [18]. However, it has been reported that ZnO NPs may exhibit cytotoxic effects in mammalian cells [19,20]. The toxicity depends on many factors, such as the concentration, size and shape of ZnO NPs, their cellular uptake, generation of reactive oxygen species (ROS) and induced inflammatory responses. Recent data showed that the cytotoxicity of PMMA/ZnO NPs composites depends on the concentrations of incorporated ZnO and the degree of its release from the composite [21]. Therefore, each new PMMA/ZnO NPs composite needs to be tested to avoid potentially adverse reactions in recipients during prolonged exposure.

New dental materials are being constantly developed to improve their physical, mechanical and functional properties. Among the latter, biocompatibility is one of the most important, where the tissue’s response to the dental materials is measured. This may influence the success of any dental treatment, especially when there is a need to have a dental material in contact with the tissue for a long time (as in the case of dentures). For cytotoxicity testing, different animal cell cultures can provide information on the basic biological behaviour of the dental material [22]. Cell cultures, such as mouse L929 fibroblasts, are useful models since they provide large amounts of consistent cells and because most cellular characteristics are maintained, which provides reliable experimental results [23].

This study aimed to synthesise new composite made of PMMA reinforced with ZnO NPs through conventional heat polymerisation. This new material is very attractive to researchers but has not appeared on the market and has no commercial application as yet. However, the method used for obtaining the composite in our work is much simpler, and therefore, more economically affordable for both the manufacturer and the patient. It was expected that a completely new microstructure and mechanical and functional properties of PMMA/ZnO NPs would be achieved with this approach. With the use of different microscopy techniques, the microstructure evaluation of the PMMA/ZnO NPs composite was performed by testing their mechanical properties. According to the specific application (in dentistry), specific attention has been given to the investigation of the functional properties, namely the measurement of colour and cytotoxic activity in the PMMA/ZnO composite, in comparison with a nontoxic control PMMA composite. The cytotoxicity investigations included direct and indirect exposure of the target cell line, L929 cells, to material and material extracts, respectively, according to the Specification Standards for Medical Devices, ISO 10993-5: 2009(E) [24]. Producing the new dental composite using PMMA as the matrix and ZnO NPs as the reinforcing elements could lead to a longer life for dental dentures, which in combination with the appropriate cytotoxicity, would be helpful for many patients.

## 2. Materials and Methods 

### 2.1. Materials 

Commercially available nanopowder, ZnO (high purity: 99.99%, density: 5.606 g/cm^3^, insoluble in water, formula weight: 81.37, melting point: 1975 °C) was purchased from Interdent (Belgrade, Serbia). The average size of the ZnO NPs was 30 nm and they were spherical. The ZnO NPs are shown as white particles mixed with PMMA powder in Figure 1, where the image was taken on a Sirion electron microscope 400 NC (Field Electron and Ion Company—FEI, Hillsboro, OR, USA). For the synthesis of the new composite based on PMMA, the use of 2 wt.% ZnO NPs was chosen, according to our previous experience and based on the literature [25].

The PMMA matrix (PMMA powder with benzoyl-peroxide, pigments and the liquid methylmethacrylate HQ 60, ethylene glycoldimethacrylate, methylmethacrylate composite) was purchased from Galenika (Belgrade, Serbia). This material was used for Biogal^®^ acrylic teeth Galenika (Belgrade, Serbia) and has already passed the appropriate ISO standards.

### 2.2. Synthesis of PMMA and PMMA/ZnO NPs Samples

The polymerisation of a new composite started when the monomer was added to the powder mixture of PMMA and ZnO NPs. The initiator generates a free radical that attacks the double bond on a monomer molecule, bonding to it and causing the combined molecule to have a free radical at the end. This reacts with another monomer molecule and the process continues until another initiator free radical terminates the chain. Typically, there is one growing chain per micelle at a time, leading to polymer particles with nearly identical molecular weights. The synthesis of the pure PMMA and PMMA/ZnO NPs composite was conducted using heat polymerisation (100 °C, t = 1 h 45 min) with the use of the pressing technique. The proper composition was selected for each type of sample, combined with the powder, polymethyl methacrylate, catalyst, pigments, liquid, methyl methacrylate stabiliser, dimethacrylate and 2 wt.% ZnO for production of the composite. The integrated dosage system was used to provide an ideal mixing ratio and minimum polymerisation shrinkage in the samples. This was then followed by mixing the powder and the liquid in the given ratio with a spatula. The mixture was left to mature in a closed mixing cup at room temperature for approximately 5–10 min. As soon as the material had matured sufficiently and was no longer sticky, it was worked further for approximately 20 min at room temperature. Next, the pressing phase was performed: A sufficient quantity of resin dough was placed in the hand-warmed (approx. 40 °C), isolated flask (cuvette) halves. The flask (cuvette) was closed carefully, loaded with an 80 bar pressure and fixed with a clamp. Polymerisation followed the standard procedure: The closed flask (cuvette) was placed in cold water, heated up to 100 °C and allowed to boil for 45 min. This was used because the residual monomer content can be reduced by increasing the polymerisation temperature. The final cooling was done by leaving the flask (cuvette) at room temperature for at least 30 min. Subsequently, we completely cooled all samples with cold water. 

It should be noted that four different types of samples were synthesised: (i) for microstructural investigation and density measurements, discs with φ = 10 mm and h = 1 mm were created; (ii) for testing mechanical properties (compressive test), real denture prostheses were made; (iii) for colour measuring, discs with φ = 13 mm and h = 1.5 mm were made, and (iv) for biocompatibility testing, samples that were 5 mm × 5 mm × 1 mm in size were created. Samples from groups (iii) and (iv) were polished in a wet condition with water and bimstein dust; a deerskin brush was used for the final fine polishing. The PMMA samples represented the control group throughout the research of the microstructure and all the above mentioned properties in comparison with the new composite PMMA/ZnO NPs.

### 2.3. Microstructure Analysis

The following equipment was used for microstructure observation: environmental scanning electron microscope (FEI Quanta 200 3D, Hillsboro, OR, USA) and a scanning electron microscope (FEI Sirion 400 NC, Hillsboro, OR, USA). The samples, after previous preparation (polishing with C-paste to about a size of 3 µm), were sprayed with Au (Jeol JSM 8310 appliance, Tokyo, Japan), which enabled the observation of the non-conductive surfaces of the samples with an electron beam. To get the best possible insight into the structure of the new composite, we observed the fracture surfaces. The fracture was performed in liquid nitrogen, thus achieving a brittle fracture and preventing ZnO NPs from falling out of the PMMA matrix. The samples coated with a thin layer of Au on the surface were positioned in the chamber of the microscope and observations were performed with an accelerating voltage of 15 kV. 

After connecting the STEM (scanning transmission electron microscopy, FEI, Hillsboro, OR, USA) detector to the FEI Quanta 200 3D microscope, we characterised samples with transmitted electrons at the nanometer level. The STEM detector had two solid-state segments that allowed for high-resolution bright- and dark-field imaging. These observations allow us to obtain information on the crystalline defects in metals, ceramics and integrated circuits, and can be applied for bacteriological studies, autoradiography for genetic studies, general observation of cellular morphology, culture samples, nanoparticles, etc. The detector can hold up to eight transmission electron microscopy (TEM) grids for samples (Agar Scientific Ltd, Stansted Essex, UK), it can be mounted easily on the stage of the microscope and it can operate with a low or high vacuum with electron accelerating voltages from 5 to 30 kV. It can act as a pre-analysis tool for TEM that involves greater electron accelerations and resolution. 

### 2.4. Compressive Testing of a Real PMMA/ZnO NPs Denture

The mechanical properties of a real PMMA/ZnO NPs denture were measured according to Standard SIST EN ISO 604 and were tested with compressive tests on a ZWICK/ROELL Z010 materials testing machine (Zwick/Roell, Fürstenfeld, Austria) at a crosshead rate of 2.0 mm min^−1^. Special supports were created (with special models of jaws) such that the models of complete dentures could be fixed into the machine, and the supports were placed in such a way as to simulate the occlusal relationship between the upper and lower teeth in the mouth. The compressive tests were supposed to establish any influence from the surface preparation procedure and resulting border surface on the tooth point, such as on the denture base due to pressure from chewing forces that dental prostheses pass on to the jaw segment during their function in the oral cavity. The real denture specimen was compressed along its major axis at a constant speed until the decrease in length reached a predetermined value. The applied load and compressive strain were recorded. The loading of the specimen was stopped at 50% compressive strain, mainly for safety reasons. The load–strain responses were used to calculate the compressive strength σ_M_. 

### 2.5. Density Measurements with a Pycnometer 

A pycnometer is a device that determines the density of a liquid or samples. It is made of glass, with a close-fitting ground glass stopper with a capillary tube through it, such that air bubbles may escape from the apparatus. First, the pycnometer is weighed empty, then is filled with a liquid whose density is known; in our case, water was used (density 1 g/cm^3^). Then, the sample is added to the pycnometer, which is filled with water and weighed. The weighing was carried out at a temperature of 20 °C. To calculate the density, we used average values of the sample mass (PMMA pure, PMMA/ZnO NPs 2%) and the average mass value of a pycnometer containing a liquid (H_2_O density = 1 g/cm^3^, ethanol density = 0.79 g/cm^3^).

### 2.6. Colour Measurements

A Datacolor SF600 Plus spectrophotometer (Konica Minolta, Osaka, Japan) was used for measuring the colour parameters of the PMMA/ZnO NPs samples (pure PMMA was used as the control). The instrument allowed for measuring the remission spectrum of incident light in 10 nm intervals within the visible part of the spectrum (360–700 nm) [26].

### 2.7. A Data Analysis

The CIELAB (known as CIE L*a*b* or abbreviated as simply “Lab” color space) color coordinates (L*, a*, and b*) were calculated from the diffuse reflection measurements for a D65 standard illuminant using CIE 1931 standard colorimetric observer data. The differences in the brightness (ΔL*), intensities, and directions of the red–green coordinate (Δa*) and yellow–blue coordinate (Δb*) were calculated relative to the colour of the reference samples (PMMA vs. PMMA/ZnO NPs). The total change in colour (ΔE*), chroma (ΔC*) and hue (ΔH*) were calculated using the following Equations:ΔE_a_ = [(ΔL*)^2^ + (Δa*)^2^ + (Δb*)^2^]^1/2^(1)
C*_a,b_ = (a_*_^2^ + b_*_^2^)^1/2^(2)
ΔC* = C*_a,b samples_ − C*_a,b reference_(3)
h_a,b_ = arctan(b*/a*)(4)
Δh_a,b_ = h_a,b samples_ − h_a,b reference_(5)
ΔH* = 2 × ( C_* samples_ - C* _reference_ )^1/2^ sin(Δh/2)(6)

### 2.8. Cytotoxicity Investigations

The cytotoxicity of the PMMA/ZnO NPs composite was evaluated in a culture with an L929 cell line (ATCC), which is recommended for the evaluation of cytotoxicity of medical devices according to the Specification Standards for Medical Devices ISO 10993-5: 2009(E) [24]. In addition to the PMMA/ZnO NPs composite, pure PMMA control samples were used for comparison in the assays. To assess how the PMMA composite affected the L929 cells directly, samples sized 5 mm × 5 mm × 1 mm were placed on the adherent L929 cell monolayer for 24 h. As a negative control, a glass coverslip with the same dimensions was used, which is a material that does not induce a cytotoxic reaction but may exhibit similar pressure-related effects on the L929 monolayer. As a positive control, a preparation known to induce cytotoxic changes was applied, which consisted of a sterile water solution of phenol concentrated to 4% on a sterile filter paper (Merck Millipore, Darmstadt, Germany) that was 5 mm in diameter. After 24 h of incubation at 37 °C in 5% CO_2_ in a complete RPMI medium (10% foetal calf serum, 1% antibiotics), the L929 monolayer was assessed using phase-contrast microscopy, especially in the place of contact between the test material and the cells. Additionally, the viability of the cells was measured using a Trypan blue exclusion assay after the collection of all cells, both dead and alive, using 0.1% trypsin in a 0.02% NaEDTA solution. To assess the indirect effect of the composites, the extracts of PMMA/ZnO NPs and PMMA samples were obtained by conditioning the samples in a complete medium for 7 days. The surface of the sample over the medium volume was 4 cm^2^/mL. The samples were vortexed for 1 min at 1200 rpm, covered with aluminium foil and placed in the incubator. After the incubation, supernatants were collected and their pH values were determined (pH510 Euthech instruments, Shanghai, China). The extracts were used as the dilutions of a control blank medium (90%, 45%, 22.5%), whereas the control was the extract of a blank medium (0%). The extract of the negative control was used as a concentrate, whereas a 5% phenol solution in the blank medium was used as a positive control. The L929 cells were cultivated in 96-well plates until reaching 80%–90% confluence, and then the sample extracts were added for the next 24 h. The cytotoxic effects were evaluated using an MTT assay, which enabled the assessment of cellular metabolic activity coming from the viable cells. After culturing with the extracts, the cultures were incubated with 3-(4,5-dimethylthiazol-2-vl)-2,5-diphenyltetrazolium bromide (MTT, 0.5 mg/mL) for 4 h, followed by treatment with 0.01 NHCL in 10% SDS/water overnight. The absorbance was read using a spectrophotometer (DC 990 BV, microtiter plate reader, NT laboratory, Roma, Italy) at 540 nm, and the reference wavelength (670 nm) was subtracted. All samples were tested six replicates. The relative viability was calculated based on the corrected optical densities, scaling blank control samples to 100%.

## 3. Results and Discussion 

### 3.1. Synthesis of Composite PMMA/ZnO NPs

We wanted to show that by simply mixing the powder of both substances (PMMA and ZnO NPs powder) in a vacuum condition, we could obtain a material with improved properties. When we started investigating this new material, there were only a few articles, and these only considered applications in engineering and other branches, not dentistry. Last year, expansion was started regarding investigations of PMMA/ZnO NPs composites in the dental fields [21,27,28] because of their good antimicrobial properties. Our research was tasked with showing the fracture resistance of new material on the real tooth PMMA/ZnO NPs denture, while other work was based on testing materials in the form of plates, discs, fibres or films. The Balkan Peninsula is still undergoing financial recovery; therefore, it was necessary to find the simplest way to produce a new composite with practical applications such that patients from this region could be offered a cheaper solution that did not require frequent repairs or replacement of the denture base and acrylic teeth.

Based on our previous research [25], it was determined that the percentages of ZnO NPs in the PMMA matrix of 2 vol.% and 3 vol.% are sufficient to improve the physical and mechanical characteristics of the newly obtained PMMA/ZnO NPs composite.

The graph that best explains the range between the minimum volume representation of the filler (area I) and the maximum (Vc—critical volume in area II) is shown in Figure 2 [29].

The mentioned volume percentages from previous studies [30] were not enough to obtain the desired properties, and a higher percentage than those used in this research would not lead to better results than those achieved, and could even lead to worse results.

### 3.2. SEM Characterisation

A microscopic examination of a fracture surface of composite PMMA/ZnO NPs confirmed the homogeneous distribution of ZnO NPs on the inspected surface, as seen in Figure 3.

The fracture showed that the PMMA/ZnO NPs composite was brittle, which is characteristic of polymer materials. A detailed observation revealed signs of depolymerised dark fields around small agglomerations of ZnO NPs. This phenomenon could lead to deteriorated mechanical properties of the composite. Consequently, it is necessary to avoid the depolymerisation process via adequate surface modification of the ZnO NPs [31]. The homogeneously dispersed ZnO NPs in the PMMA matrix could be explained using the theory of different kinds of integration by grafting copolymer chains [32,33]. Some authors concluded that the planarity of composite PMMA with some other particles (silica, TiO_2_) can be significantly improved by using surface modification [34]. A simulation of the surface distribution of ZnO NPs on a PMMA matrix was prepared with the use of Energy Dispersive X-ray (EDX) analysis and module X-mapping (Figure 4). The simulation showed a good surface distribution of ZnO NPs, while the calculated theoretical value of ZnO NPs was 2.2 wt.%. The higher content of simulated wt.% value was probably due to the computer processing of the places where the chemical elements Zn and O were detected using EDX analysis. This indicates a process error of X-mapping since the wt.% of ZnO NPs was lower. Regardless, we can summarise that, through conventional heat polymerisation, it was possible to synthesise homogeneous composite PMMA/ZnO NPs. 

The EDX mapping showed a relatively homogeneous distribution of ZnO NPs on the PMMA surface, with no apparent major clustering of the ZnO NPs in the PMMA matrix. The distribution is visible in Figure 3 at low, medium and high magnifications, with corresponding magnification scales. The image capture resolution was 512 × 384 pixels of width and height. Each red point represents a pixel where Zn was identified on the surface of the sample. The corresponding EDX spectrum showed Au content, along with C, O and Zn from the PMMA/ZnO NPs composite, which was a result of Au coating of the samples for SEM observations. The electron acceleration voltage for acquiring the EDX analysis was 10 kV.

### 3.3. Compressive Testing 

The results of the compressive testing are presented in Table 1, where a big difference was found in the forces for the first break that were applied to complete dentures made of pure PMMA and composite PMMA/ZnO NPs. The first crack was noticed on the display of the Zwick Roell machine and the denture as well, as shown by the arrow in Figure 5. A more than five-fold bigger force was necessary for the lower complete dentures and almost two-fold larger force for the upper complete dentures made using new composite PMMA/ZnO NPs instead of pure PMMA under identical conditions. These results should be connected with the average masticatory forces in the posterior bite area in jaws. Most authors suggest that a functional force of 500 N is a physiological maximum in natural dentition [35,36,37]. Some of them considered that masticatory forces vary in a range from 50 to 250 N during normal mastication, but can reach over 500 N if there is some parafunction, such as bruxism [38].

### 3.4. Density

The calculated density values are presented in Table 2. Although there were small differences in the specific density values in the composite PMMA/ZnO NPs in comparison with the pure PMMA, a tendency toward increasing density was noticed, which could be indirectly connected with the incorporation of ZnO NPs on the free places in the long PMMA chains during the polymerisation process. Those results are in accordance with other results found using the Flash DSC (Differential scanning calorimetry) and DMA (Dynamic mechanical analysis) methods, from which we can conclude that polymerisations were carried out to the end, and thus, the presence of a residual monomer was reduced to a minimum or does not exist. When there is no residual monomer, the main cause of allergic reactions, then the risk of their occurrence is reduced. Furthermore, there are no trapped spaces in the PMMA matrix, which would be an ideal place for the accumulation of food, bacteria and fungi that could cause other oral manifestations. Those results are also followed by increased E modulus and damping and reduced level of cross-linking, which leads to this material being more friendly to oral mucosa due to vibration damping. This phenomenon consequently resulted in fewer residual monomers and is in concordance with some previous research [25,31,39]. Consequently, PMMA/ZnO NPs present dentistry and patients with a new option for PMMA replacement because it is expected, due to its higher density, to have a reduced risk of allergic reactions; specifically, allergic reactions are closely connected to the presentation of residual monomers and porosity inside of polymer materials [22], which could lead to lower mechanical properties too.

### 3.5. Colour 

Adding ZnO NPs to PMMA gave a brighter colour since the percentage of nanoparticles increased (Figure 6); this was expected since it is known that ZnO is used to obtain the white colour used in painting, known as a zinc white [40]. Table 3 provides the different values for all parameters that define colour (L, a and B), and from that, we could conclude that the average value for B was 2.7 times higher in the new composite than in pure PMMA. The PMMA/ZnO NPs composite had a brighter colour than the pure PMMA (∆E = 6.2). This problem could be solved in the future by adding a proper pigment, which would result in a similar colour to the teeth or oral mucosa, which are replicated in resin teeth and denture bases.

### 3.6. Cytotoxicity 

The cytotoxicity tests were carried out using a direct contact between the test material and L929 cells; the tests showed that the PMMA samples did not exhibit cytotoxic effects in culture. This was concluded based on the unaltered morphology of the L929 cells and the undisrupted cobblestone-like morphology of the L929 cell monolayer. In contrast, in culture with positive control samples (phenol solution of filter paper), a completely disrupted cell morphology and no adherent cells were present after 24 h culture, as expected (Figure 7). These results were confirmed using the Trypan blue exclusion test, showing that the percentage of dead (Trypan blue positive) cells was similar in the negative control, glass-covered, PMMA and PMMA/ZnO NP samples (Table 4).

To assess whether the PMMA samples exhibited any indirect effect via released products, the extracts of PMMA and PMMA/ZnO NP samples were prepared in a cell culture medium for 7 days, followed by the cultivation of L929 cells in this medium (Figure 8). The results from the MTT assay suggested that extracts from the PMMA and PMMA-ZnO NPs samples in the highest concentration used (90% in a complete medium), did not reduce the viability of L929 cells more than 10% compared to the untreated control, which according to the ISO 10993-5: 2009(E) [24] Standard, could be interpreted as the lack of any cytotoxic effect. Other extracts and dilutions, in contrast to the positive control, did not affect the viability of L929 cells either. Therefore, both qualitative and quantitative cytotoxicity tests carried out in direct and indirect contact with L929 cells suggested that the investigated samples (PMMA/ZnO NPs and PMMA) did not cause cytotoxicity. The absence of cytotoxicity of our samples of PMMA is in accordance with published results [41]. In contrast, ZnO NPs are cytotoxic [19,20,42]. The finding that the samples of PMMA/ZnO NPs did not cause cytotoxicity is in agreement with some recent studies, where PMMA/ZnO NPs composites were prepared differently. A study on the L6 myoblast cell line showed that a PMMA/ZnO NPs composite was non-cytotoxic (the viability was >85%), but the content of incorporated ZnO NPs was only 0.1% (*w*/*v*) [28], in contrast with our composite, where the concentration of ZnO was 2%. When ZnO NPs (2.5%) were incorporated into PMMA, the concentration of the released ZnO was 2.281 mg/L, which is far less than the cytotoxic concentration of ZnO NPs observed in that study on HeLa cells (higher than 20 mg/L) [27]. Although the authors did not test the cytotoxicity of such a prepared material, the general conclusion that a PMMA/ZnO NPs nanocomposite is non-cytotoxic could be the same.

## 4. Conclusions

In summary, we have reported the successful synthesis of a PMMA/ZnO NPs composite. The microstructure characterisation revealed the homogeneous distribution of ZnO NPs throughout the volume of the PMMA matrix. The compressive testing showed that a larger force was necessary before a fracture occurred for both the lower and upper complete dentures made using the new composite PMMA/ZnO NPs instead of pure PMMA under identical conditions. This finding needs additional investigations to confirm whether this is attributed to the ZnO NPs. The density measurements identified small differences in the values of specific density in the composite PMMA/ZnO NPs in comparison to the pure PMMA, and the higher density of the composite could be attributed to the presence of ZnO NPs. Adding ZnO NPs to the mixture with PMMA gave a brighter colour as the percentage of nanoparticles was increased. The somewhat brighter colour could be repaired with pigments, similar to oral tissues. Based on our investigation, it could be concluded that the addition of ZnO NPs into a matrix of PMMA gave a new composite material with proven biocompatibility, i.e., it has no cytotoxic activity on L929 cells.

## Figures and Tables

**Figure 1 materials-13-02717-f001:**
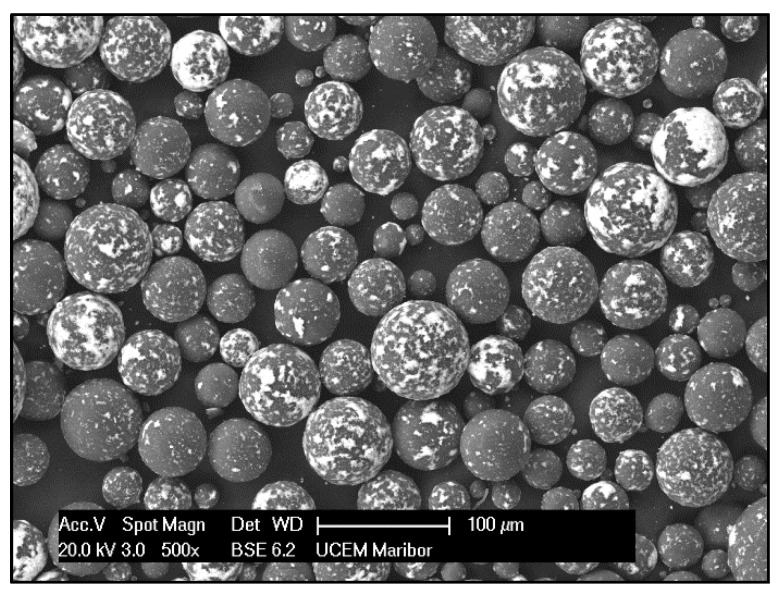
Scanning electron microscopy (SEM) micrograph of poly(methyl methacrylate) (PMMA) matrix powder (grey spheres) mixed with ZnO nanoparticles (NPs) (white nanoparticles fixed on the surface of grey spheres).

**Figure 2 materials-13-02717-f002:**
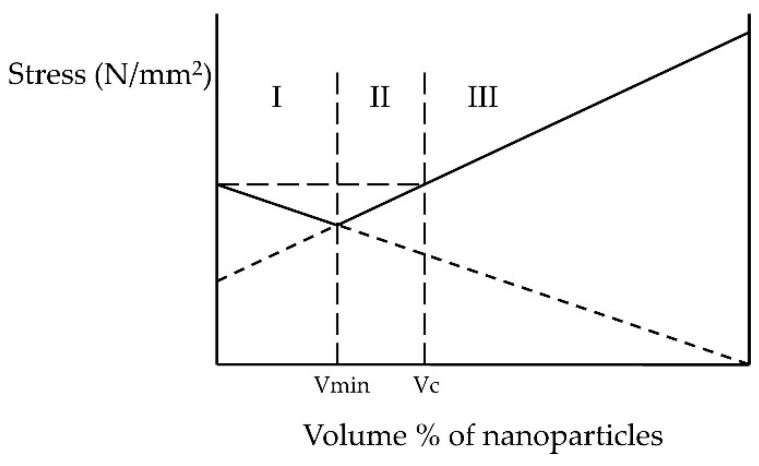
Behaviour of materials with different proportions of nanoparticles in the matrix; Vmin—minimal volume % of nanoparticles, area I—unchanged properties of the material, Vkr—maximum volume % of nanoparticles, area II—improved properties of the material, area III—possible deterioration of properties.

**Figure 3 materials-13-02717-f003:**
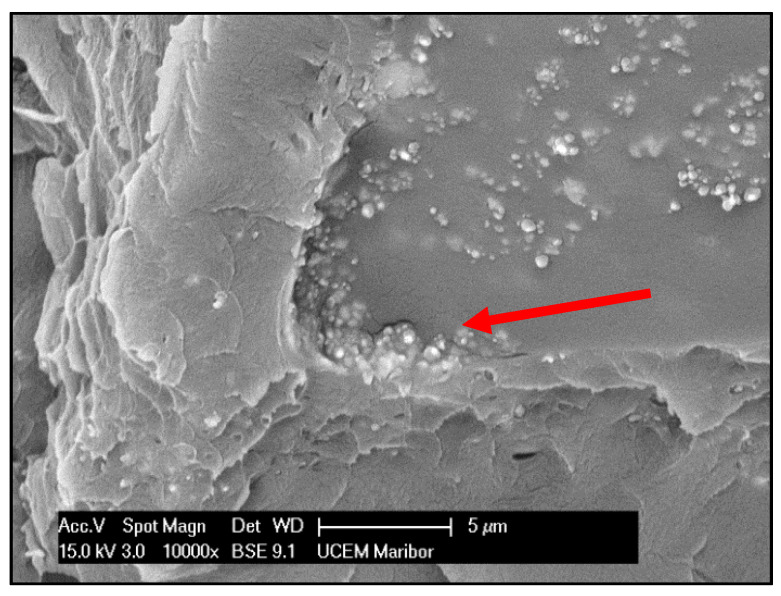
SEM micrograph of the PMMA/ZnO NPs composite’s microstructure showing a fractured surface. ZnO NPs are visible as white particles, while the depolymerised dark field is shown by the red arrow.

**Figure 4 materials-13-02717-f004:**
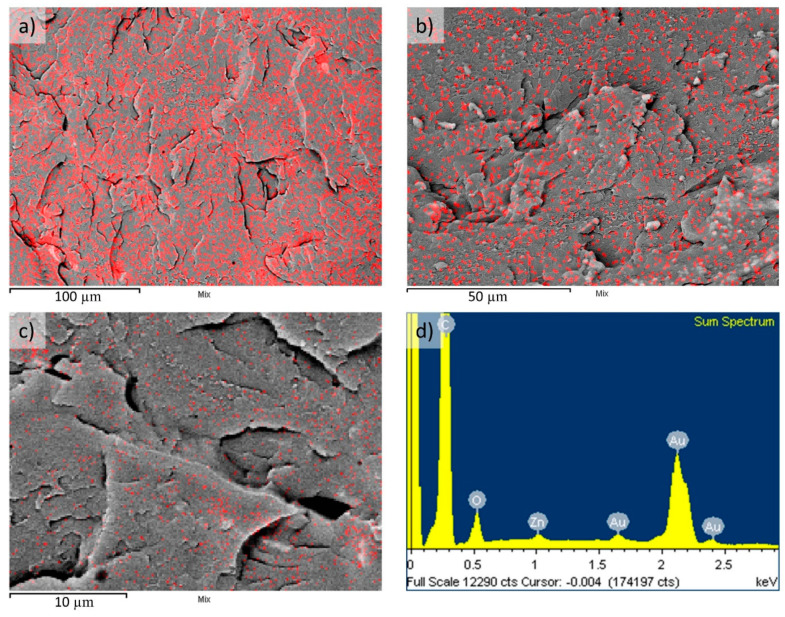
Surface distribution of ZnO NPs on a PMMA matrix (visible as red points) from EDX mapping with low (**a**), medium (**b**) and high magnification (**c**), with the corresponding EDX spectrum from the analysis (**d**).

**Figure 5 materials-13-02717-f005:**
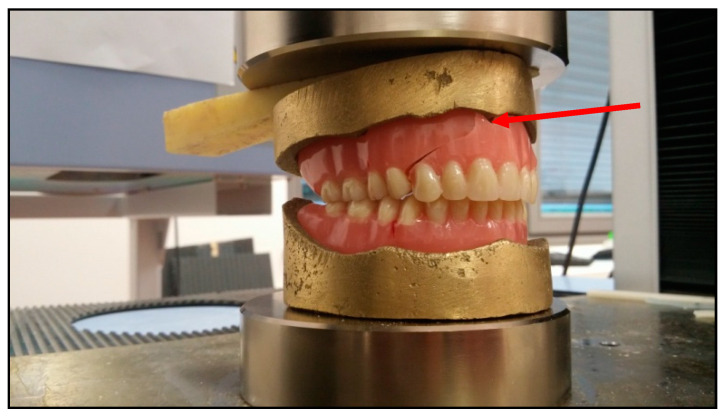
Tooth denture PMMA/ZnO NPs after the compressive testing and the place where the crack started (red arrow).

**Figure 6 materials-13-02717-f006:**
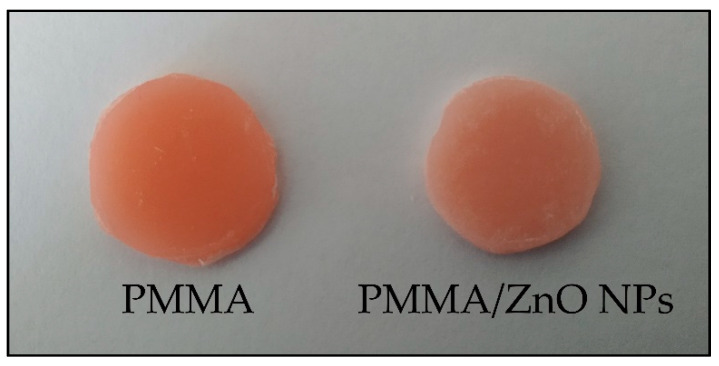
Presentation of the PMMA and nanocomposite PMMA/ZnO NPs (photo).

**Figure 7 materials-13-02717-f007:**
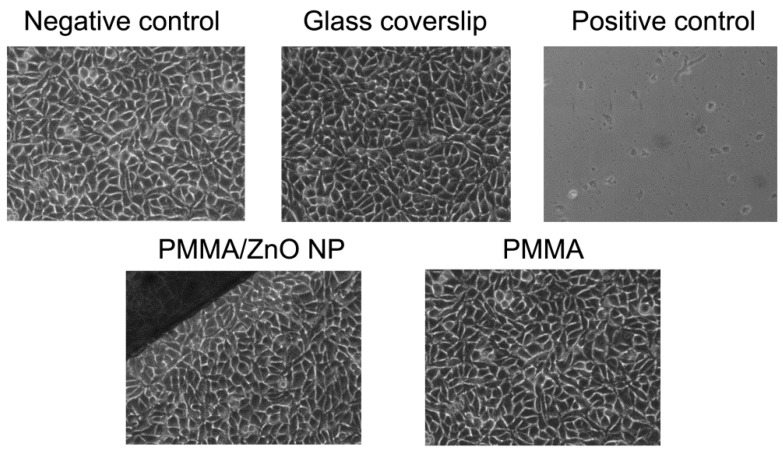
Phase-contrast analysis of L929 cells in direct contact with PMMA samples. Images of the L929 cell cultures were taken after 24 h cultures in direct contact with PMMA, PMMA/ZnO NP, a glass coverslip, a positive control sample or blank cell cultures, at a magnification 10× using a phase-contrast microscope.

**Figure 8 materials-13-02717-f008:**
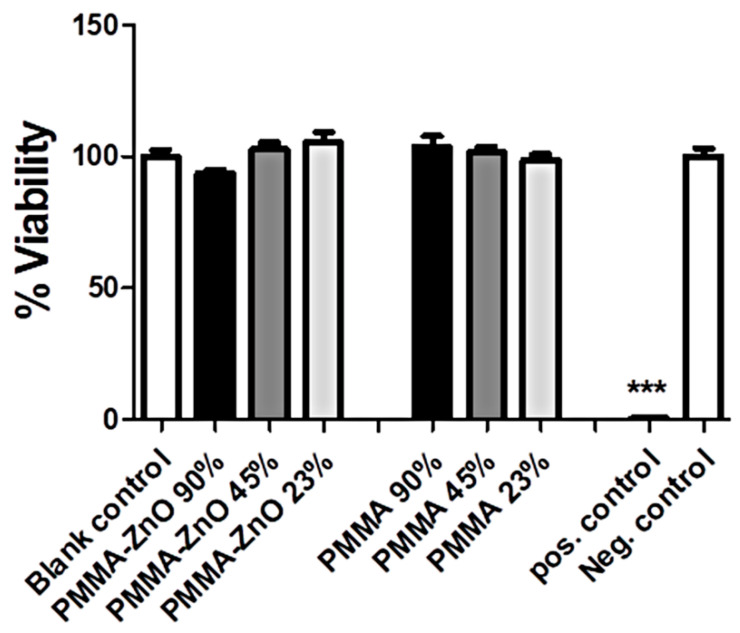
Indirect effect of PMMA sample extracts on the viability of L929 cells. The PMMA and PMMA/ZnO extracts were prepared by conditioning the samples in a complete RPMI medium for 7 days, as described in the Materials and Methods section.

**Table 1 materials-13-02717-t001:** Results of the compressive testing of a real tooth PMMA/ZnO NPs denture.

Real Tooth Denture Samples	Force for the First Break in the Lower Complete Dentures (N)	SD	Force for the First Break in the Upper Complete Dentures (N)	SD
PMMA	451	32	2983	167
PMMA/ZnO NPs	2266	156	4000	201

**Table 2 materials-13-02717-t002:** Results of calculated density.

Density (g/cm^3^)	PMMA Pure (g)	PMMA/ZnO NPs (g)	Pycnometer (g)	Pycnometer + H_2_O (g)	Pycnometer + Ethanol (g)
Average	1.2204	1.2371	40.9941	90.9341	81.4159
Min.	1.2202	1.2266	40.9940	90.9266	81.3887
Max.	1.2207	1.2444	40.9942	90.9427	81.4443
STDEV	0.0002	0.0075	0.0001	0.0075	0.0219

**Table 3 materials-13-02717-t003:** Results of the colour measurements.

Sample	L	A	B	ΔE
PMMA	94.8	17.5	9.8	-
PMMA	96.7	17.7	10.0	
PMMA	93.5	17.5	10.1	
PMMA average	95.0	17.6	10.0	
PMMA/ZnO NPs	89.6	18.5	12.7	6.2
PMMA/ZnO NPs	89.4	18.6	12.6	
PMMA/ZnO NPs	89.7	18.7	12.7	
PMMA/ZnO NPs average	89.6	18.6	12.7	

**Table 4 materials-13-02717-t004:** Quantitative cytotoxicity in direct contact with L929 cells.

Percentage of Dead L929 Cells				
Type of Sample	Sample 1	Sample 2	Sample 3	Mean Value
Positive control (phenol solution)	100%	100%	100%	100%
Negative control (Saligal, dental pump)	2%	1%	1%	1.33%
Negative control (cells alone)	1%	2%	2%	1.67%
PMMA/ZnO NPs composite	1%	1%	2%	1.33%
PMMA	2%	2%	1%	1.67%

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
