# Peer review of "Microstructure Characterisation and Identification of the Mechanical and Functional Properties of a New PMMA-ZnO Composite"

_materials, 2020, doi:10.3390/ma13122717_

Round 1

Reviewer 1 Report

The manuscript "Microstructure Characterisation and Identification of the Mechanical and Functional Properties of A New PMMA-ZnO Composite written by Rebeka Rudolf et al., describes a synthesis and basic study of new PMMA-ZnO material. The manuscript is well written and results are adequately described. I have only few comments:

  1. The novelty of the manuscript needs to be better highlighted in the introduction part as well as in the discussion. There are many similar materials and it has to be clearly defined, where this one excells.
  2. Results on the figure 3 should be better described. What parameters of the simulation were used and what information is contained in the figure?

Author Response

1 reviewer:

  1. The novelty of the manuscript needs to be better highlighted in the introduction part as well as in the discussion. There are many similar materials and it has to be clearly defined, where this one excells.

This new material is very attractive to researchers, but has not yet appeared on the market and has no commercial application. However, there is a difference in the method of obtaining a composite in our work, that is much simpler and, therefore, more economically affordable for both the technique and the patient.

  1. Results on the figure 3 should be better described. What parameters of the simulation were used and what information is contained in the figure?

The results have been elaborated in the added text below Figure 3, with an explanation of the information in the Figure and parameters of the analysis. The Figure was corrected with added images of the ZnO NP distribution on the PMMA surface, in low, medium, and high magnifications, with corresponding magnification scales.

Reviewer 2 Report

In this manuscript, the authors prepared PMMA-ZnO nanocomposite materials for medical usage, considering it is improved mechanical properties and good biocompatibility. The experimental results do show the improve compressive property and comparable non-cytotoxicity. However, there are several questions in the manuscript below:

  1. The studies related to PMMA-ZnO nanocomposite are very popular. I didn’t see any novelty for this study. The mechanical and biocompatible properties are widely discussed in others’ work. I don’t think it is the first time to report it. Please demonstrate the novelties of this study or any differences. I could easily find the similar studies:

https://doi.org/10.1016/j.apsusc.2016.05.122

https://doi.org/10.3390/nano9091318

https://doi.org/10.1016/j.msec.2019.03.053

  1. In Figure 1, the scale bar is 100 um, suggesting that the ZnO is not nano-sized, which is conflicted with Figure 2.

  1. The EDS mapping is a must to verify the distribution of ZnO in the matrix. The overall and zoom-in mapping are both required.

  1. The optical images of prepared samples with different amounts of ZnO are required to evaluate the color of it directly.

  1. Are there other mechanical properties critical for dental materials such as elastic modules?

Without answering these questions, the manuscript cannot be considered to publish. 

Author Response

2 reviewer:

  1. The studies related to PMMA-ZnO nanocomposite are very popular. I didn’t see any novelty for this study. The mechanical and biocompatible properties are widely discussed in others’ work. I don’t think it is the first time to report it. Please demonstrate the novelties of this study or any differences. I could easily find the similar studies:

https://doi.org/10.1016/j.apsusc.2016.05.122

https://doi.org/10.3390/nano9091318

https://doi.org/10.1016/j.msec.2019.03.053

We wanted to prove that by simply mixing the powder of both substances (PMMA and ZnO NPs powder) in a vacuum condition - to obtain a material (PMMA-ZnO) with improved properties. In the moment we started investigation on this new material, there were only a few articles, but only with application in Engineering and other branches, except Dental. Last year expansion in investigations of PMMA/ZnO NPs composites was started on the dental fields [references 21,26,27], because of their good antimicrobial properties. Our research was tasked with proving the resistance of the new PMMA-ZnO composite on concrete denture samples for fracture resistance, while other work was based on testing materials in the form of plates, discs, fibres or films.  The Balkan Peninsula is still a scene of financial recovery, and it was necessary to find the simplest way to produce a new composite with concrete application in practice, so that patients from this region could be offered a cheaper solution which will not require frequent repairs or replacement of the denture base and acrylic teeth.

We cited some additional references in Introduction, including the suggested ones about cytotoxicity of ZnO NPs and the mechanisms involved (see page 2). In addition, we compared in the section Cytotoxicity the effect of PMMA/ZnO nanocomposites published in other studies, using different models, with ours (see page 10 and 11).

  1. In Figure 1, the scale bar is 100 um, suggesting that the ZnO is not nano-sized, which is conflicted with Figure 2.

There was an error in the manuscript. Figure 1 actually represents PMMA powder (grey spheres) mixed with ZnO NPs (white particles). Corrections have been made in the Figure 1 description and in the text above the Figure. 

  1. The EDS mapping is a must to verify the distribution of ZnO in the matrix. The overall and zoom-in mapping are both required.

The distribution was checked at several magnifications. The Figure now includes images with low, medium and high magnifications, with corresponding magnification scales of 100, 50 and 10 µm. This gives the reader a possibly better overview of the overall and zoomed-in distribution of the particles in the PMMA matrix. The image capture resolution was 512x384 pixels, as given in the added paragraph below the Figure, for a more detailed explanation of the EDX mapping results.

  1. The optical images of prepared samples with different amounts of ZnO are required to evaluate the color of it directly.

We added optical images (Figure 6), where a lighter shade of conventional PMMA colour is observed. Of course, as a higher percentage of ZnO nanoparticles is added to the PMMA matrix the colour is much lighter, since it is known that ZnO is also used to obtain the white colour used in painting.

  1. Are there other mechanical properties critical for dental materials such as elastic modules?

We investigated elastic modules in our previous work with DMA and the Flash DSC technique [24], and found out that ZnO NPs added to the PMMA increased the E modulus and dumping, and reduced the level of cross linking, which leads to this material being more friendly to oral mucosa due to vibration damping.

Reviewer 3 Report

New dental materials are being developed constantly in order to improve their physical, mechanical and functional properties. This research work presented the synthesis of Polymethyl-methacrylate (PMMA) enriched with 2 wt. % of zinc-oxide nanoparticles (ZnO) through conventional heat polymerization and the characterization. The PMMA-ZnO composite exhibited better mechanical performance. This work has the potential for practical application, I recommend the publication of this manuscript after addressing some questions, as follows:
1. The authors mentioned, “the 2 wt.% of ZnO NPs was chosen, according to our previous experiences and based on literature.” Please add more discussions for choosing 2 wt.% of ZnO NPs.
2. Our diet often contains acidic foods like carbonated drinks or apricots. Whether the new dental composite based on ZnO NPs as the reinforced elements is resistant to acid corrosion?
3. In part 3.3, the authors mentioned, “PMMA/ZnO NPs present for Dentistry and patients one of the new options for PMMA replacement, because it is expected, due to higher density, to have reduced risk of allergic reactions”. Since the average density difference between PMMA/ZnO NPs and PMMA is rather limited, only 0.016g, is this statement appropriate? Please add more discussions on it.

Author Response

3 reviewer:

New dental materials are being developed constantly in order to improve their physical, mechanical and functional properties. This research work presented the synthesis of Polymethyl-methacrylate (PMMA) enriched with 2 wt. % of zinc-oxide nanoparticles (ZnO) through conventional heat polymerization and the characterization. The PMMA-ZnO composite exhibited better mechanical performance. This work has the potential for practical application, I recommend the publication of this manuscript after addressing some questions, as follows:

  1. The authors mentioned, “the 2 wt.% of ZnO NPs was chosen, according to our previous experiences and based on literature.” Please add more discussions for choosing 2 wt.% of ZnO NPs.

Based on our previous research [24], it was determined that the percentages of ZnO NPs in the PMMA matrix of 2 vol.% and 3 vol.% are sufficient to improve the physical and mechanical characteristics of the newly obtained PMMA/ZnO NPs` composite. The graph that best explains the range between the minimum volume representation of the filler and the maximum (critical volume) is shown in Figure 2 [27] and added in the text. Previous studies [28] were not enough to get the desired properties, and a higher percentage than those used in this research would not lead to better results than those achieved, while it could even lead to worsening.

  1. Our diet often contains acidic foods like carbonated drinks or apricots. Whether the new dental composite based on ZnO NPs as the reinforced elements is resistant to acid corrosion?

We are currently on a project where we are testing corrosion on a different dental materials, in PMMA/ZnO NPs also, so we still do not have published results to the public.

  1. In part 3.3, the authors mentioned, “PMMA/ZnO NPs present for Dentistry and patients one of the new options for PMMA replacement, because it is expected, due to higher density, to have reduced risk of allergic reactions”. Since the average density difference between PMMA/ZnO NPs and PMMA is rather limited, only 0.016g, is this statement appropriate? Please add more discussions on it.

In the PhD thesis [38] the author pointed that even though this density increased by only 0,016 this result had a positive trend with others results from the Flash DSC method [24], from which we can conclude that polymerisation were carried out to the end, and, thus, the presence of residual monomer is reduced to a minimum, or does not exist. When there is no residual monomer, the main cause of allergic reactions, then the risk of their occurrence is reduced. Also, there are no trapped spaces in the PMMA matrix, which would be an ideal place for the accumulation of food, bacteria and fungi that can develop other oral manifestations.

Round 2

Reviewer 2 Report

I reviewed the updated version of the manuscript. It is much better than the original one. The questions have been carefully answered. I'd like to approve its publication.